# Exploring the Role of Land Transfer and Social Capital in Improving Agricultural Income under the Background of Rural Revitalization

**DOI:** 10.3390/ijerph192417077

**Published:** 2022-12-19

**Authors:** Haiyan Yu, Wenjie Zhang, Shuai Pang

**Affiliations:** 1School of Emergency Management, Xihua University, Chengdu 611730, China; 2Sichuan Academy of Social Science, Southwestern University of Finance and Economics, Chengdu 611730, China; 3School of Economics, Yunnan University, Kunming 650091, China; 4College of Economics, Sichuan Agricultural University, Chengdu 611100, China

**Keywords:** social capital, farmland transfer, agricultural income, rural China

## Abstract

Under the background of rural revitalization in China, with the process of urbanization and the implementation of China’s land system reform, rural workers gradually gain multiple income streams. However, increasing agricultural income remains the final guarantee for professional farmers to shake off poverty, and land is still their last security. We applied the OLS model and mediation model to a dataset of 3789 households in 25 provinces obtained from China Family Panel Studies (CFPS) to investigate the influence of farmland transfer and social capital on farmers’ agricultural incomes. The results show that farmland “transfer in” and social capital significantly help to increase agricultural income directly, and farmland “transfer in” behavior plays a vital mediating role, influencing the positive effect of social capital on agricultural income. The study examined the logical social capital-agricultural land transfer-agricultural income correlation in the progression of rural society, from “hollow” to “reflux”, under the continuous expansion of rural revitalization strategies, which is of great practical significance for re-recognizing the positive role of rural social capital and agricultural land transfer in improving the income of professional farmers and realizing the overall goal of rural revitalization. The results also provide a theoretical basis for guiding and leveraging the effective use of social capital to promote agricultural land transfer.

## 1. Introduction

In China, the Rural Revitalization Strategy was first formally proposed in the report to the 19th National Congress of the Communist Party of China, aiming to address the issues concerning agriculture, rural areas, and farmers in a high-quality manner, to expand middle-income groups and to achieve common prosperity for all people. The issue of absolute poverty in China was solved historically; however, there will still be people earning low incomes in society. According to the *Bulletin of the Third National Agricultural Census*, by the end of 2016, large-scale businesses accounted for only 1.92% of the total agricultural households registered in China [1]. For professional farmers, agriculture is the main source of income. In the modern era, the key to consolidating the achievement of poverty alleviation is to effectively and consistently increase the income of farmers who live in rural areas. Due to productivity disadvantages, how to increase agricultural income has become key to promoting common prosperity to achieve substantial progress, as well as being a highly regarded research topic.

On the one hand, social capital is essential to improve farmers’ lives, which is the essence of rural revitalization [2]. Traditional rural China is a society of acquaintances or semi-acquaintances [3], giving rise to social capital, which is an invisible social resource based on emotional connections. This relationship network, composed of blood relationships, geography, and kinship, is the main form of rural social capital in China [4]. In China, personal relationships are extremely important in society; thus, social capital exists among family, kinship, and neighborhood relationships. Therefore, it is widely believed that social efficiency can be improved through the network relations, norms, and trust that comprise social capital [5]. Especially in rural areas, where market systems may not yet be perfected, this traditional “Chinese guanxi” plays an important role in resource allocation, such as creating employment opportunities, increasing the income of the most disadvantaged, narrowing the income gap, and mitigating risk impact, which all play an immeasurable role in reducing the incidence of poverty [6]. In poor rural areas, where both physical capital and human capital resources are relatively scarce, social capital has played the role of “poor capital” to a certain extent [7]: helping individuals to obtain sufficient market information, reducing transaction costs and promoting cooperation by virtue of private trust and interpersonal relations [8]; easing financial pressure through mutual borrowing between relatives and friends, filling financial loopholes in rural areas as an informal capital carrier [9], reducing the credit constraints of farmers by representing informal guarantees [10], and providing a sustained impetus to reduce or even eliminate long-term poverty [11]. Social capital in the form of potential capital (such as the help of relatives and friends) also plays a good role in resisting risk [12]. Moreover, social capital can also indirectly improve the level of income by influencing investments in human capital and material capital [13].

In the contemporary period, social capital is still playing a role in promoting farmers’ income. Under rural revitalization strategies and land system reforms, the mobility of rural residents has greatly increased, leading to gradual loosening of the interpersonal network and reshaping traditional rural social capital which is mainly characterized by acquaintances and closed networks; thus, the countryside is no longer a closed community [14]. In the process of agricultural modernization, gradual improvements in the market system and the development of rural areas from closed to open are widely believed to have eroded the “Chinese guanxi” basis of the role of traditional rural social capital, which has led to doubts about the role of social capital in promoting farmers’ income. For example, Zhang Shuang et al. hypothesized that with improvements in marketization, the role of social capital in poverty will be weakened, and especially the role of family social networks [15]. However, it should also be observed that the urban-rural relationship has smoothly undergone the rural “hollowing out” stage and entered the return stage of migrant workers, which has brought back talent, technology, and capital. At this time, the network of new social capital is more open and contains a wider range of social relationships, including more urban relationships. On the normative side, the digital economy and the Internet have prompted social capital to act in a more resilient manner. Additionally, the Internet has further reduced constraints on the level of trust, improved the gray character which traditional social capital possesses, and expanded the possible ways in which social capital can work to boost farmers’ incomes and fight poverty. Under rural revitalization, novel rural social capital is becoming more modern, advanced, extensive, and digital. As a kind of invisible “soft” capital, it plays a wider role in attracting human capital, material, and financial resources, forming a good circulation network between rural and urban areas, and reinforcing mechanisms of the role of traditional social capital in promoting farmers’ agricultural income, i.e., the expansion of farmers’ financial sources, the enrichment of information, and the establishment of social networks. Therefore, social capital and rural revitalization are mutually supportive. The embedding of social capital is conducive to the realization of rural revitalization. The implementation of the rural revitalization strategy enriches rural social capital and attracts more social capital back to the countryside [2]. Moreover, the degree of modernization in rural areas in China is still weak in the transition period. Under the condition that “system” and “market” are not sound, “relationship” plays an extremely important role as one of the bases and ways for farmers to obtain scarce resources [16].

Under rural revitalization strategies, local employment remains the most effective and sustainable means of alleviating poverty, considering the long-term effectiveness and cost. The cyclical opening and closing of cities caused by the COVID-19 pandemic has led to a decline in employment demand in some urban industries, especially in the contact service industry, ultimately leading to a substantial increase in migrant workers returning home for employment. Data from the Ministry of Agriculture and Rural Affairs of the People’s Republic of China showed that by the end of 2021, 11.2 million entrepreneurs had returned to the countryside, 1.1 million more than in 2020 [17]. Therefore, the local employment of migrant workers urgently needs to promote the large-scale development of agricultural industrialization. Land is the most basic and important production factor for agricultural-scale management; moderate-scale agriculture relies on the consolidation of agricultural land and farmland transfer [18]. Thus, an increase in land transfer provides an important guarantee for agricultural modernization [19]. In rural China, the *Household Responsibility System,* implemented in the early 1980s, promoted production motivation in peasants, and led to serious farmland fragmentation, which curbed large-scale agriculture operations [20]. In 1988, the transfer of land contract management rights was officially permitted [21], which effectively promoted the transfer of farmland and created better conditions for moderate-scale agricultural management. By the end of 2019, the degree of farmland transfer exceeded one-third of the total farmland area [22]. For professional farmers, one of the best ways to increase income is by transferring into land and developing it into a large agricultural enterprise. In order to improve the efficiency of resource allocation, professional farmers should expand the scale of farmland to cater to contemporary agricultural production [23].

Thus far, connections between social capital and agricultural development have been the key focus in academia, such as the associations between different forms of social capital and innovation in agriculture [24], social capital’s critical role in agricultural and rural development in various countries [25], as well as the role of social capital in development of agricultural entrepreneurship [26]. However, the existing research literature is fragmented with respect to the impact of social capital on farmers’ income and the impact of land transfer on agricultural scale management, ignoring the important intermediary role of land transfer. From the perspective of traditional human social relations, studies on how to increase agricultural income in the modern era are insufficient; the importance of land to farmers is often ignored. Therefore, it is of great practical significance to study the influence of new social capital for rural revitalization on agricultural income, through embedding into land transfer.

Land transfer is measured by two variables: land “transfer in” and land “transfer out”. Land “transfer in” means that farmers rent land from other local farmers through written or verbal contract [27]. It is more directly related to farmers’ agricultural income, and was the research object of this study. China is a society of traditional “Chinese guanxi”, in which social capital plays a supplemental role in the formal system [28]. In fact, rural households with more capital can more easily transfer into land for large-scale production, and thus earn more farm income [29]. Land transfer is an important bridge for social capital to embed in rural revitalization, which ultimately raises farming income for farmers.

As an intangible capital embedded in the vast rural areas, social capital plays an important role in achieving regional agricultural specialization and scale development, as well as supplementing agricultural modernization. The trust and relationship values of social capital are embedded in the process of land transfer and become informal institutions which influence people’s interactive behavior, thus improving the financial capacity of individuals or households to transfer into land on the one hand, and increasing the possibility of transferring into land through social relationship networks on the other. Therefore, with continuous rural revitalization, it is highly significant to explore the logical correlation between social capital, farmland “transfer in”, and agricultural income with the contemporary rural society becoming more open, which is important for reconceptualizing rural social capital, improving professional farmers’ income, and promoting the overall goal of rural revitalization.

The key research questions of this study are:(1)How do land “transfer in” and new social capital affect agricultural income under rural revitalization?(2)Is farmland “transfer in” one of the important ways for the new social capital to influence agricultural income?(3)Under the constraint of the scale of farmland “transfer in”, is there a nonlinear relationship between social capital and agricultural income?

## 2. Materials and Methods

### 2.1. The Effect of Social Capital on Agricultural Income

Farmers are the largest social group in China, but also a group with relatively low income, lower level of living security, and weaker endowments of market resources. In the foreseeable future, farmers will still be heavily reliant on the countryside and agriculture, and those who relocate to cities may still return the countryside [3]. Agricultural income refers to the minimum earnings for farmers who cannot live in cities and those who have returned to their hometowns to maintain basic living security. Social capital can effectively reduce the incidence of poverty and the income gap through a variety of ways, as has been confirmed by many scholars [2,6,11,13]. Thus, there is also a direct impact of social capital on the agricultural income of general agricultural workers, who are the largest population among the most disadvantaged in society. For most villagers who have lived in the countryside for a long time, their traditional social capital has been seriously drained with farmers’ urbanization. Although rural revitalization has injected modern features into the social capital and reconstructed the rural social structure, characterized by a traditional closed-loop, this has led to an irreplaceable role of promoting farmers’ agricultural income using new social capital.

The report to the 19th National Congress of the Communist Party of China clearly put forward the general requirements for the implementation of rural revitalization from five aspects. These were correlated to contemporary rural issues, mainly referring to revitalizing the rural industrial economy, optimizing the rural living environment, reconstructing rural local culture, promoting effective governance and improving farmers’ living standards [30]. The massive backflow of rural labor has brought advanced production factors such as urban technology, knowledge, experience, and capital to rural society with the support of the rural revitalization policy. Additionally, those strategic resources were effectively diffused through the relationship network of social capital, on the one hand alleviating the loss of traditional rural social capital and the weak social capital of long-staying rural farmers, mainly women, the elderly, and children, and sustaining traditional rural social capital in continually increasing farmers’ income. On the other hand, urban capital is injected into rural capital with the return of migrant rural workers, alleviating the plight of farmers who find it difficult to raise funds to expand agricultural production, especially in remote mountainous areas in China, where the returned social capital becomes a key resource. At the same time, as an information carrier, novel social capital implies a broader social relationship referring to urban relations, carrying advanced production factors which can improve agricultural productivity and modernization, and ultimately contributing to increased farmers’ agricultural income.

Under rural revitalization, as an invisible “soft” capital, new rural social capital has a broader role in attracting human capital, material and financial resources, forming a good circulation network between rural and urban, and reinforcing the mechanisms of the role of traditional social capital in promoting farmers’ agricultural income, notably through the expansion of farmers’ financial sources, the enrichment of information and the establishment of social networks.

Based on the above discussion, this study proposes the first research Hypothesis:

**Hypothesis 1 (H1).** 
*Rural social capital can still contribute to the growth of agricultural income of farm households through various mechanisms, i.e., social capital has a direct relationship with agricultural income.*


### 2.2. Influence of Land “Transfer in” Behavior on Farmers’ Agricultural Income

As a typical “Chinese guanxi” society, the characteristics of “acquaintance society” exist across rural China. Social relations play a very important role in resource acquisition, distribution, and transaction [31], and become an important influencing factor in the farmland transfer. Among the forms of land transfer, the most popular is land leasing. Farmers pay a certain amount of farmland rent and lease land, thus expanding their production scale. When the marginal revenue exceeds the marginal cost, farmers can profit from agricultural production. The income-increasing effect of agricultural land transfer is mainly manifested in the economy of scale effects, economy of scope effects, and the labor division effects of agricultural production. Farmland “transfer in” may exert several effects on agricultural income. The first and most important effect is the scale effect, then the scope economy effect, and the division of labor effect. Specifically, the transfer of agricultural land has realized increases in land input factors, and expanded the production boundary of land factors under the actions of advanced technology, knowledge, and human and material capital of rural revitalization; thus, increases in land inputs can bring increasing returns to scale and promote the effective expansion of agricultural output. Specifically, land “transfer in” implies a land input increase for agricultural production, whereas the improved technology, knowledge, talent, and material capital of rural revitalization extend the production boundary of land factors, enabling increases in land inputs to bring incremental returns to scale and promote the effective expansion of agricultural output. Then, the expansion of cultivated land area enlarges the adjustable range of agricultural production [32]. The average long-term cost of output tends to decline as the category of agricultural output increases. By optimizing the planting structure and factor input ratios, farmers can achieve the best agricultural outputs. Finally, the “transfer in” of farmland is conducive to the moderate-scale operation of agriculture, and the consolidation of farmland is conducive to the mechanization of agriculture [33]. These are conducive to the growth of new agricultural business entities led by professional farmers and of family farms. At the same time, however, some farmers who “transfer out” of the land will change from professional farmers to part-time farmers or enter the non-agricultural population. This differentiation of the labor force will undoubtedly lead to a differentiation between agricultural production and income. Accordingly, research Hypothesis 2 was formulated:

**Hypothesis 2 (H2).** 
*Land “transfer in” can promote farmers’ farm income.*


### 2.3. Impacts of Social Capital on Farmland “Transfer in” Behaviors

The role of social capital in all aspects of agricultural production is similar. The process of land transfer for agricultural income generation involves reduced transaction costs, the access to lease capital and information, risk sharing and cooperation mechanisms in the establishment of long-term land transfer behavior, and the input of urban production factors, all carried out under a network of rural social interactions. Therefore, rural social capital can effectively promote farmers’ land “transfer in” behavior, which, in turn, promotes agricultural income growth. Thus, land transfer plays an intermediary role in the process of social capital promoting the growth of farmers’ income and is one of the mechanisms through which social capital directly affects farmers’ agricultural income. Accordingly, research Hypothesis 3 was formulated:

**Hypothesis 3 (H3).** 
*Social capital has a positive effect on land “transfer in”, and land “transfer in” is an important way for social capital to increase agricultural income, and plays an intermediary role.*


Finally, existing studies have identified that the more disadvantaged people are not originally superior in terms of social capital accumulation, i.e., social capital does not significantly increase agricultural income when farmers’ social capital is low, and social capital can only significantly contribute to higher farm household income and thus reduce farm incomes poverty when social capital crosses a certain threshold value [34]. Therefore, research Hypothesis 4 was:

**Hypothesis 4 (H4).** 
*The scale of rural land “transfer in” has threshold characteristics. Under the constraints of different scales of farmland “transfer in”, the impact of social capital on agricultural income is also different.*


The theoretical analysis framework of “social capital→farmland transfer in→agricultural income” and hypotheses are shown in Figure 1.

## 3. Data, Variables and Methodology

### 3.1. Data

The data used in this study were acquired from China Family Panel Studies (CFPS). According to the introduction from http://isss.pku.edu.cn/cfps/index.htm (accessed on 30 August 2022), China Family Panel Studies (CFPS) is a nationally representative, annual longitudinal survey of Chinese communities, families, and individuals launched in 2010 by the Institute of Social Science Survey (ISSS) of Peking University, China. The survey focuses on economic activities, education, family behaviors, etc., and covers 25 provinces (cities and autonomous regions) in China, over a wide range of locations, giving it great research value. It is presented in the form of questionnaires, including individual questionnaires, family questionnaires and village questionnaires.

This study used data obtained in 2020, which are the most recent publicly available data. Some village-level control variables were used in this study; therefore, the CFPS 2014 village data and 2020 household data were matched and combined. In addition, some variables from the 2010 baseline survey which were not included in CFPS 2020 were used. The research object was agricultural income; thus, the samples of households not engaged in agricultural production were omitted. After processing, this study obtained a final 3789 effective samples of agricultural families, which were distributed in 297 villages around 25 provinces (municipalities directly under the Central Government and autonomous regions) in China. Compared with previous studies, the data used in this study have the advantages of a large sample size, new data, and wide geographical coverage.

### 3.2. Variables

#### 3.2.1. Dependent Variable

The dependent variable in this study was the agricultural income of households engaged in agricultural production. The Family Questionnaire of CFPS 2020 includes the question “In the past 12 months, how much does your family get by selling agricultural products, including the crops you cultivated, forestry products, poultry, livestock, fishery products and other sideline products (for example, eggs, piglets, etc.) produced or raised by your family?” The answer to this question was the dependent variable we adopted in this study. The unit of measurement of the variable was CNY, and we treated the data logarithmically.

#### 3.2.2. Main Independent Variable

##### Farmers’ Social Capital

To date, due to the extensive meaning of social capital, there is still no unified standard for selecting indicators for this variable. According to the definitions of social capital outlined by James S. Coleman et al. [35], all points emphasize the connections among people based on reciprocity and trustworthiness. Generally, the maintenance of this kind of connection not only depends on blood relationships and geographical connections, but also on the economic exchanges. To reflect the level of favor exchanges engaged in by farmers, gift spending is a stable index [36]. In addition, the total amount of money spent annually on relatives, neighbors, and friends, including transfers, represents the daily economic exchanges [28]. Communication fees were also considered in this study.

In the family questionnaire of CFPS 2020, all the answers pertaining to communication fees, gift spending, and money given to relatives and to others were totaled to measure the level of favor exchanges, which represents the spending on social capital. The unit of measurement of this variable was CNY, and we treated these data logarithmically.

##### Land “Transfer in”

Although there are many different forms of land transfer in rural China, in the questionnaire, only farmland leasing was studied. Thus, in this paper, land “transfer in” refers to renting in farmland from others. In the family questionnaire, one question is “In the past 12 months, did your family rent any other land than the collectively distributed land from other people or the village collective, regardless of paying land rent or not?”. The answer to this question represented the independent variable.

Land “transfer in” here was a dummy variable: if the farmer had rented farmland from others in the past year, the variable value was 1; otherwise, the value was 0.

#### 3.2.3. Control Variable

Inspired by previous studies, in order to control the impacts of other factors on agricultural income, this study controlled the characteristics of family and village. In addition, province dummies were included.

Control variables relating to the family included farmland size, family size, farm machinery rental fee, family deposits, machine values, other income sources (including wage income, individual operation income, and transfer income), and life status (including water type, toilet type, and house type).

Village control variables included village economic status, altitude, village farmland size, the number of noticeboards, and urban and rural categories.

The model variables are all described in Table 1.

### 3.3. Methodology

To evaluate the influence of social capital and land “transfer in” on agricultural income, the benchmark model was set to Equation (1).
*Agricultural income = β_0_+ β_1_ Social capital +β_2_ Land transfer in + γX + ε*(1)

To test the intermediary effect of land “transfer in”, referring to Baron and Kenny [37], a three-step regression was used in this study. Three regression models were set, as Equations (2)–(4).
*Agricultural income = α_0_ + α_1_ Social capital + γX + ε*(2)
*Land transfer in = θ_0_ + θ_1_ Social capital + γX + ε*(3)
*Agricultural income = β_0_ + β_1_ Social capital +β_2_ Land transfer in + γX + ε*(4)

To test H4, whether the scale of rural land transfer exhibits threshold characteristics, the threshold regression model was set to Equation (5).
*Agricultural income = φ_0_ +φ_1_ Social capital ·I(Scale of land transfer in ≤λ) + φ_2_ Social capital ·I(Scale of land transfer in > λ) + γX + ε*(5)

In Equations (1), (2) and (4), agricultural income is a continuous variable, and these three models adopted multiple linear regressions; in Equation (3), the dependent variable of land “transfer in” is a binary variable, which uses the Probit regression model; in Equation (5), the threshold variable of the scale of land “transfer in” equals the ratio of farmland “transfer in” area to the households’ operational farmland acreage, where *I*(·) is the indicator function and *λ* is the threshold value, *X* represents a series of matrix of control variables, *ε* is a standard error, and *α, β, φ, θ,* and *γ* are the parameters to be estimated.

## 4. Empirical Results

### 4.1. The Impact of Social Capital and Land ‘Transfer in’ on Agricultural Income

Stepwise regression was adopted to estimate the effect of social capital and land “transfer in” on agricultural income. In addition, in this study, the proxy variable of social capital was the money spent on maintaining the “acquaintance relationship”, which could lead to a two-way causal relationship. Thus, the variable of social capital was an endogenous variable in the basic model. To solve this problem, an instrumental variable was adopted. In the village questionnaire of CFPS 2014, one question concerns the “Proportion of popular surname”. In rural China, in the same village, people with the same surname are usually more closely related. Residents with the same surname can represent more useful social capital for villagers in their daily life. It is obvious that “the proportion of popular surname” is a good instrumental variable, which is not only exogenous, but also highly correlated with the independent variable. After the exogenous test and weak identification test, the instrumental variable was shown to be effective. Then, 2SLS and GMM models were used to estimate the impact of social capital on agricultural income.

Table 2 reports the empirical results of the OLS, 2SLS and GMM models. Models (1) to (5) show the step-by-step addition of control variables; the results of Models (6) and (7) were obtained after adding instrumental variables.

Table 2 shows that the coefficients of social capital and land “transfer in” are consistently significantly positive in the process of adding control variables one by one. The estimation results presented in Table 2 indicate that more social capital can help farmers access more effective help and increase agricultural income. At the same time, the “transfer in” of agricultural land is conducive to the expansion of the scale of agricultural production, which can help increase agricultural income. The results support research Hypotheses 1 and 2 of this study. Social capital effectively promotes the development of rural society through network, norm, trust, and other attributes. Farmers can have advantages in agricultural management and production by virtue of the materials and relationship support provided by social capital; thus, the amount of social capital that farmers have is transformed into their ability to obtain agricultural income. Similarly, the increasing input of land factors can still significantly promote agricultural production in the process of rural modernization. Social capital can form a positive interaction with land “transfer in”, and better social capital can promote the smooth conclusion of land leasing, whereas land leasing can effectively expand interpersonal networks and trust capital, which together can contribute to the increase in farmers’ agricultural income.

When household and village characteristics are controlled, the results of model 5 show that the effects of housing type, wage income and urban households on agricultural income are negative but insignificant, whereas the effects of the remaining control variables are positive, but not all significant. The type of house reflects the economic level of farmers to a certain extent; therefore, it will affect their agricultural income. People with wage-earning and urban households tend to have less willingness to engage in agriculture, i.e., income diversity can replace agricultural income and lead to a decrease in farm income. Models (6) and (7) imply that, after dealing with endogenous problems effectively, the coefficients of the main independent variables become larger.

### 4.2. Robustness Test

Subsequently, we used the method of throwing samples to examine the robustness of the benchmark regression results. Firstly, Model (8) reports the impact of social capital and farmland “transfer in” on agricultural income with the whole sample. Secondly, as a comparison, Model (9) to Model (12) report the regression results of the partial samples. The research object of this study was households engaged in agricultural production which were mainly located in rural areas; urban households were included in Model (9). Similarly, village samples without cultivated land were removed in Model (10). In Model (11), the agricultural households’ samples without contracted farmland were removed, because their agricultural outputs were all dependent on land leasing, which may interfere with the regression results. Finally, because China’s agricultural production is based on households, and small-scale operations still occupy a large proportion, the samples of family farms with a farmland acreage larger than 30 Mu is excluded in Model (12) to test whether the effect of social capital on agricultural income is stable among farmers engaged in small-scale agricultural production.

Table 3 reports the parameter estimation results with the change in sample size. With the decrease in sample size, the influence of social capital and farmland “transfer in” on farmers’ agricultural income is still significantly positive. For farmers whose farmland operation scale is less than 30 Mu, the contribution of social capital to agricultural income is more obvious, and the marginal contribution is up to 27.92%. Thus, for professional farmers who stay in rural areas for a long time, possessing a certain amount of social capital can provide them with certain resources for agricultural production. Rural social capital undergoing revitalization still actively influences farmers’ willingness and behaviors. It can be seen from Table 3 that the results of the benchmark regression model are still reliable when the sample size changes.

### 4.3. Individual and Regional Heterogeneous Impacts

This study adopted quantile regression (QR) to estimate the individual heterogeneity in Table 4. For different quantiles of agricultural income, impacts of social capital and farmland “transfer in” on agricultural income were different. Model (13) and Model (14) show parameter estimates for selected quantiles (25%, 50% and 75%). Model (14) deals with the endogenous problem. Model (14) implies that, with the increase in quantile, the coefficient of social capital increases significantly. In other words, compared with farmers with low agricultural incomes, for farmers with higher agricultural incomes, social capital plays a greater role in increasing agricultural income. However, the effect of land “transfer in” on agricultural income is the opposite. Model (14) shows that for farmers with low agricultural incomes, land “transfer in” plays a more important role in increasing agricultural income, which, to some extent, implies that the agricultural-scale operation of low-income farmers is insufficient, and increasing the land use area is more relevant for low-income farmers to increase their agricultural income.

To evaluate regional heterogeneity, rural China was separated into two parts. Model (15) outputs the regression result of the main-grain producing area in China. In contrast, Model (16) presents the regression result of other regions in China. In the 13 main grain-producing areas, the output value of the primary industry accounts for a large proportion, and the grain output accounts for more than 70% of the national share. Due to different geographical factors, the types of agricultural crops in the main-grain producing area and other regions are different, and the degree of agricultural modernization is also different. Comparing the results of Model (15) and Model (16), we can see that social capital has a significant positive effect on agricultural income in the main-grain producing area. However, in other regions, the role of social capital is positive, but not significant. Farmland “transfer in” is helpful to increase agricultural income, but this effect is more obvious in secondary grain-producing areas. This result suggests that the use value of land should be further reflected in rural areas and farmers’ income space should be expanded and fully integrated with local conditions.

### 4.4. The Intermediary Mechanism Test

In order to further determine the mechanisms of social capital and farmland “transfer in” on farmers’ agricultural income, this study took farmland “transfer in” as an intermediary variable and further estimated Equations (2)–(4). After performing the Sobel test, the original hypothesis that there was no mediating effect was rejected, and a mediating effect was shown to exist. Table 5 presents the three-step regression results. Model (17) shows that social capital has a significant positive effect on agricultural land “transfer in” at a significance level of 5%, indicating that the role of social capital is conducive to the “transfer in” of farmland and the expansion of agricultural scale. Model (18) is the direct impact of social capital on agricultural income. Model (19) shows that the effect of social capital on the increase in agricultural income is significantly weakened after adding the intermediary variable. The regression coefficient changed from 0.1818 to 0.1656, which further verified the existence of partially mediating effects. This conclusion proves that the “transfer in” of farmland is an important mechanism for social capital increasing agricultural income, and plays an intermediary role.

### 4.5. The Threshold Mechanism Test

The effect of social capital on agricultural income is strongly dependent on land “transfer in”, which implies that there is a threshold of the scale of land “transfer in” to be identified. After attempting to identify the threshold effects, we obtained statistically significant results with three stable threshold values. As shown in Table 6, the first and second threshold values of the farmland “transfer in” scale were 0.14 and 0.176, respectively, at a significance level of 5%. The third threshold value of the farmland “transfer in” scale was 1.5 with a significance level of 10%. Table 7 presents the results of the threshold regression. The results in Table 7 imply that under the constraint of different scales of farmland “transfer in”, the influences of social capital on agricultural income are disparate. Under the condition that other related variables are controlled, it can be found that when the ratio of farmland “transfer in” area to the households’ operation farmland acreage is ≤14%, the effect of social capital on agricultural income is significantly positive, but when the ratios of farmland “transfer in” area to the households’ operation farmland acreage are > 14% and ≤17.6%, the effects of social capital on agricultural income are negative. When the ratio of the farmland “transfer in” area to the households’ operation farmland acreage is >17.6%, the role of social capital will significantly contribute to the increase in agricultural income, and with the expansion of land “transfer in” scale, this role will become more obvious. This result also provides strong evidence for moderate-scale agricultural operations.

## 5. Discussion

Based on the data of 3789 rural households from the China Family Panel Study, this study examined the direct impact of farmland “transfer in” and social capital on agricultural income, and the mediating role of land “transfer in” in the process of social capital contributing to farm income generation. Compared with previous studies, marginal contributions of this study include: (1) Based on the era of rural revitalization, this study explored the impact of new social capital on the agricultural income of farmers who have stayed in rural areas for a long time on the basis of traditional rural social capital; this study not only considered the direct impact of social capital and land “transfer in” on agricultural income, but also tested whether land “transfer in” behavior plays a mediating role in the effect of social capital on agricultural income, establishing a theoretical analysis framework of “social capital→farmland transfer in→agricultural income”. (2) This study responds to current questions about whether the new social capital reconstructed under China’s rural revitalization strategy can still significantly contribute to farmers’ agricultural income, and confirms that social capital still plays an active role in China’s new rural areas. (3) This study has dealt with the endogenous problem, and discussed the individual heterogeneity of farmers and regional differences in rural China. (4) This study has discussed the nonlinear relationship between social capital and agricultural income under the constraints of different scales of farmland “transfer in”, which enriches the literature on agricultural income. (5) From a worldwide perspective, the use of the “acquaintance relationship” is beneficial to farmers, which has implications for all agricultural households.

For farmers, land is the most basic living guarantee, and agricultural income is the most basic source of income. How to increase agricultural income is a topic worth considering under rural revitalization strategies. Currently, the weakest links to achieving widespread prosperity are agriculture, rural populations, and farmers. For rural areas in the primary stage of revitalization, it is still necessary to strongly consider social capital. However, the use of the “acquaintance relationship” should only be a supplement to the standardized market, and should not become the mainstream.

There are many similarities and differences between this study and previous analyses presented in the literature. Social capital contributes to farmland transfer, which is similar to the results obtained by of Chen, H. and Wang, J. [38]; however, we further analyzed the complex relationship among social capital, land “transfer in”, and agricultural income. Moreover, based on the novel background of the gradual disintegration of traditional rural society, this study explored whether new social capital under rural revitalization can still promote growth in agricultural income, and addressed some doubts. Qian [39] studied the role of social capital in land transfer; however, we discussed the endogeneity and heterogeneity, expanding the research future. Regarding the relationship between agricultural land transfer and agricultural income, this study proves once again that the moderate-scale operation of agricultural land is conducive to increasing agricultural income, which is consistent with the results of Yan et al. [40].

Our research also has some shortcomings. First of all, there were some limitations in the selection of proxy variables of social capital, which cannot summarize all the characteristics of social capital. Secondly, the impacts of social capital and farmland transfer on agricultural income are dynamic, although cross-sectional data were used in this study; thus, panel data could be used for future research. Additionally, social capital will not only affect farmland “transfer in”, but also affect farmland “transfer out”, which will also affect agricultural income; thus, the influence of social capital on land “transfer out” and agricultural income could be a research topic in future studies.

## 6. Conclusions

Based on this analysis, several conclusions can be drawn.

(1)In rural China, social capital is a useful resource for farmers, which is conducive to farmland “transfer in”, thus contributing to increases in agricultural income. With the increases in social capital investment, agricultural income will also increase, and after controlling other related variables, the marginal contribution of social capital investment to agricultural income is 16.56% and the marginal effect of farmland “transfer in” on agricultural income is 1.19. Social capital and land “transfer in” are directly related to agricultural income. In addition, the impact of social capital and farmland “transfer in” on agricultural income presents individual heterogeneity and regional heterogeneity.(2)The farmland “transfer in” is a mediating variable, which is one of the important channels for social capital to influence agricultural income. Land is a direct factor of agricultural income and one of the mechanisms by which other variables affect agricultural income.(3)Farmland “transfer in” ratio is a threshold variable, and under the constraint of the scale of farmland “transfer in”, a nonlinear relationship exists between social capital and agricultural income.(4)Farmland “transfer in” is conducive to increasing agricultural income, but this effect is more obvious in secondary grain-producing areas.

From the results of this study, it seems that for farmers in an “acquaintance society”, proper maintenance of acquaintance relationship is beneficial. For government departments, the implementation of the Separation of Three Rights land policy could better address the issues concerning agriculture, rural areas, and farmers, and have a beneficial impact on local farmers’ income, complemented by social capital. Finally, the standardized development of the land transfer market is very important. Social capital plays an important role in the vast rural areas of China where market modernization is inadequate; however, the use of an “acquaintanceship” can only be a supplement to standardized markets, and the process of agricultural and rural modernization still needs to be accelerated, as well as the need for traditional social capital to be further reconstructed.

## Figures and Tables

**Figure 1 ijerph-19-17077-f001:**
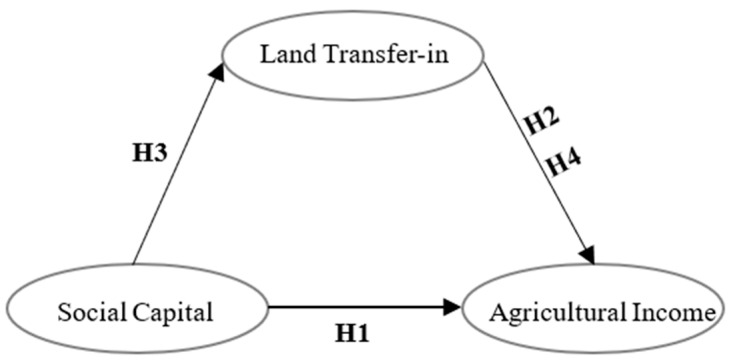
Hypothesized conceptual model.

**Table 1 ijerph-19-17077-t001:** The definition and data description of the variables.

Variable	Definition	Mean	S.D.
Agricultural income	Agricultural income (yuan)(log)	5.9121	4.6027
Social Capital	Sum of gift spending, communication fees, and money transfer to relatives and others (yuan)(log)	8.5965	1.1630
Transfer in	Whether the rural households have land “transfer in” (1 = Yes; 0 = No)	0.1494	0.3566
Farmland acreage	The total farmland area (mu) (log)	1.5779	1.1202
Machine spending	Farm machinery rental fee (yuan) (log)	3.0013	3.3545
Deposits	Total household deposits (yuan) (log)	6.6324	4.4461
Machine value	Total value of large machinery owned by family (yuan)(log)	3.4620	4.0684
Individual operation	Whether anyone in the farmer’s family engaged in self-employment (1 = Yes; 0 = No)	0.0701	0.2554
Wage or not	Whether any of the family members have a salary last year (1 = Yes; 0 = No)	0.7406	0.4383
Transfer income or not	Whether received any transfer payments last year (including government subsidies and money from others)(1 = Yes; 0 = No)	0.8258	0.3794
House type	What is the type of farmers’ house (1 = Apartment; 2 = Bungalow; 3 = Quadrangle courtyard; 4 = Villa; 5 = Condominium villa; 6 = Low-rise house; 7 = Other)	3.3484	1.9583
Water type	What kind of water does the family normally use for cooking (1 = River/Lake water; 2 = Spring water; 3 = Tap water; 4 = Mineral/Purified/Filtered water; 5 = Rain water; 6 = Cellar water; 7 = Pond water; 8 = other)	3.0018	1.2179
Toilet type	What kind of restroom/toilet facilities does the family have (1 = Indoor flush toilet; 2 = Outdoor private flush toilet; 3 = Outdoor public flush toilet; 4 = Indoor non-flush toilet; 5 = Outdoor private non-flush toilet; 6 = Outdoor public non-flush toilet; 7 = other)	4.5540	1.4808
Family size	Number of family members	4.2588	1.9468
Altitude	Altitude of the village (meter) (log)	3.7252	2.3512
Village farmland	The total farmland acreage of the village (mu) (log)	8.0946	1.4734
Notice board	Number of noticeboards in the village	2.2791	3.5452
Economic status	Visitors’ subjective evaluation of the village’s economic situation (very poor 1–2–3–4–5–6–7 very rich)	4.4549	1.3642
Urban18	Urban rural classification based on the data of National Bureau of Statistics (1 = urban; 0 = village)	0.2251	0.4177

Note: During the study period, 1 USD was equal to 6.89 RMB; 1 mu ≈666.67 m^2^.

**Table 2 ijerph-19-17077-t002:** Linear regression model results of the influence of social capital on agricultural income.

	(1)	(2)	(3)	(4)	(5)	(6)	(7)
	OLS	OLS	OLS	OLS	OLS	2SLS	GMM
Social capital	0.2777 ***	0.2425 ***	0.1533 ***	0.1585 **	0.1656 ***	4.2977 **	4.2946 **
	(0.0562)	(0.0559)	(0.0574)	(0.0630)	(0.0632)	(1.6778)	(1.6877)
Transfer in	1.5323 ***	1.3483 ***	1.2111 ***	1.2111 ***	1.1908 ***	0.7015 **	0.7012 **
	(0.1813)	(0.1799)	(0.1806)	(0.1808)	(0.1793)	(0.3181)	(0.3173)
Farmland acreage		0.5214 ***	0.4545 ***	0.4341 ***	0.4504 ***	0.4168 ***	0.4168 ***
		(0.0677)	(0.0666)	(0.0670)	(0.0664)	(0.0927)	(0.0927)
Machine spending		0.1290 ***	0.1371 ***	0.1364 ***	0.1199 ***	0.0058	0.0059
		(0.0218)	(0.0217)	(0.0217)	(0.0224)	(0.0579)	(0.0583)
Deposits			0.0720 ***	0.0716 ***	0.0703 ***	−0.0424	−0.0422
			(0.0147)	(0.0148)	(0.0148)	(0.0495)	(0.0500)
House type			−0.0191	−0.0179	−0.0216	−0.1183 *	−0.1181 *
			(0.0360)	(0.0360)	(0.0363)	(0.0649)	(0.0653)
Machine value			0.0909 ***	0.0903 ***	0.0912 ***	−0.0490	−0.0489
			(0.0168)	(0.0167)	(0.0167)	(0.0631)	(0.0633)
Individual operation				0.3274	0.3224	−1.9365 **	−1.9336 *
				(0.2633)	(0.2615)	(0.9781)	(0.9867)
Wage or not				−0.2258	−0.2302	2.2152 ***	2.2141 ***
				(0.1575)	(0.1578)	(0.8380)	(0.8411)
Transfer income				0.2262	0.2002	0.8599 **	0.8606 **
				(0.1747)	(0.1748)	(0.3788)	(0.3769)
Water kind				0.0466	0.0517	0.0703	0.0703
				(0.0580)	(0.0586)	(0.0759)	(0.0759)
Toilet kind				0.0618	0.0516	0.2102 **	0.2101 **
				(0.0568)	(0.0569)	(0.1063)	(0.1064)
Family size				0.0305	0.0265	−0.5738 **	−0.5734 **
				(0.0364)	(0.0365)	(0.2497)	(0.2509)
Altitude					0.1295 ***	0.2085 ***	0.2086 ***
					(0.0388)	(0.0633)	(0.0632)
Noticeboard					0.0721 ***	0.0397	0.0397
					(0.0210)	(0.0304)	(0.0303)
Economic status					0.1392 ***	0.2584 ***	0.2585 ***
					(0.0500)	(0.0869)	(0.0867)
Village farmland					0.0039	−0.0360	−0.0359
					(0.0493)	(0.0763)	(0.0764)
Urban18					−0.1370	0.1659	0.1669
					(0.1639)	(0.2676)	(0.2661)
Constant	6.4462 ***	5.9879 ***	5.9392 ***	5.0617 ***	6.3242 ***	27.8807 **	28.1751 **
	(0.5843)	(0.5546)	(0.5731)	(0.6989)	(0.9083)	(14.1274)	(13.0303)
Province dummies	Yes	Yes	Yes	Yes	Yes	Yes	Yes
Instrumental variable	No	No	No	No	No	Yes	Yes
R-squared	0.139	0.166	0.179	0.181	0.188		
Observations	3789	3789	3789	3789	3789	3789	3789

Note: Standard errors in parentheses; * *p* < 0.1, ** *p* < 0.05, *** *p* < 0.01.

**Table 3 ijerph-19-17077-t003:** Robustness Test.

	(8)	(9)	(10)	(11)	(12)
	Whole Samples	Located in Rural Areas	The Village Farmland Acreage > 0	Own Contracted Farmland Acreage > 0	Operation Acreage < 30
Social capital	0.1656 ***	0.2469 ***	0.2466 ***	0.2787 ***	0.2792 ***
	(0.0632)	(0.0742)	(0.0742)	(0.0778)	(0.0794)
Transfer in	1.1908 ***	1.1564 ***	1.1325 ***	1.0039 ***	1.0362 ***
	(0.1793)	(0.2064)	(0.2065)	(0.2173)	(0.2178)
Constant	6.3242 ***	5.2771 ***	4.0356 ***	4.2005 ***	4.1091 ***
	(0.9083)	(0.9483)	(0.9861)	(1.0610)	(1.0837)
Control variables	Yes	Yes	Yes	Yes	Yes
Province dummies	Yes	Yes	Yes	Yes	Yes
Observations	3789	2936	2911	2655	2508
R-squared	0.188	0.192	0.197	0.190	0.199

Note: Standard errors in parentheses; *** *p* < 0.01.

**Table 4 ijerph-19-17077-t004:** Discussions on individual heterogeneity and regional difference.

	(13)			(14)			(15)	(16)
	Q25	Q50	Q75	Q25	Q50	Q75	Main Grain Producing Area	Non Main Grain Producing Area
Social capital	0.0244	0.1291 ***	0.1785 ***	3.7157 ***	3.8134 ***	4.3397 ***	0.1614 **	0.1515
	(0.0311)	(0.0399)	(0.0359)	(0.0056)	(0.0051)	(0.0060)	(0.0798)	(0.1021)
Transfer in	0.5326 **	0.8419 ***	0.7223 ***	1.1176 ***	0.8445 **	0.3362	0.8066 ***	1.6423 ***
	(0.2513)	(0.1107)	(0.0767)	(0.3455)	(0.3171)	(0.3702)	(0.2221)	(0.2933)
Constant	9.0008 ***	6.7256 ***	7.2194 ***	−25.4701 ***	−26.8354 ***	−27.8550 ***	8.4188 ***	0.5906
	(0.6639)	(0.6682)	(0.5673)	(7.3635)	(6.7587)	(7.8907)	(1.3075)	(1.3885)
Control variables	Yes	Yes	Yes	Yes	Yes	Yes	Yes	Yes
Province dummies	Yes	Yes	Yes	Yes	Yes	Yes	Yes	Yes
Instrumental variable	No	No	No	Yes	Yes	Yes	No	No
Observations	3789	3789	3789	3789	3789	3789	2373	1416
Pseudo R^2^	0.1540	0.1371	0.0899	-	-	-	-	-

Note: Bootstrap standard errors in parentheses of model (8) and (9); Standard errors in parentheses of model (10) and (11); ** *p* < 0.05, *** *p* < 0.01. The main grain producing provinces include: Liaoning, Hebei, Shandong, Jilin, Inner Mongolia, Jiangxi, Hunan, Sichuan, Henan, Hubei, Jiangsu, Anhui and Heilongjiang.

**Table 5 ijerph-19-17077-t005:** Intermediating effect test.

	(17)	(18)	(19)
	Transfer In	Agricultural Income	Agricultural Income
Transfer in			1.1908 ***
			(0.1793)
Social capital	0.0692 **	0.1818 ***	0.1656 ***
	(0.0300)	(0.0631)	(0.0632)
Constant	−1.4550	6.5583 ***	6.3242 ***
	(0.9263)	(1.1196)	(0.9083)
Observations	3788	3789	3789
Other control variables	Yes	Yes	Yes
Province dummies	Yes	Yes	Yes

Note: Standard errors in parentheses; ** *p* < 0.05, *** *p* < 0.01. The dependent variable. in Model (17) is land “transfer in”; The dependent variable of Model (18) and (19) is agricultural income.

**Table 6 ijerph-19-17077-t006:** Threshold mechanism test.

	Threshold Value	F-Value	*p*-Value	Bootstrap Number	Critical Value		
					1%	5%	10%
Single	0.140	5.251 **	0.017	300	7.386	3.480	2.410
Double	0.176	3.358 **	0.036	2000	5.799	2.809	1.485
Triple	1.500	3.264 *	0.064	2000	6.425	3.699	2.519

Note: * *p* < 0.1, ** *p* < 0.05,.

**Table 7 ijerph-19-17077-t007:** Threshold regression results.

Independent Variable	Coefficient	Standard Errors
Social capital (Scale of farmland transfer in ≤ 0.140)	0.1886 ***	(0.0614)
Social capital (0.140 < Scale of farmland transfer in ≤ 0.176)	−0.0469	(0.0980)
Social capital (0.176 < Scale of farmland transfer in ≤ 1.5)	0.2197 ***	(0.0640)
Social capital (Scale of farmland transfer in >1.5	0.2403 ***	(0.0837)
Land transfer in	1.0821 ***	(0.1840)
Constant	−0.5360	(3.8688)
Province dummies	Yes	
Other control variables	Yes	
Observations	3789	
R-squared	0.1940	

Note: Standard errors in parentheses; *** *p* < 0.01.

## Data Availability

The data used in this study comes from China Family Panel Studies (CFPS) (http://isss.pku.edu.cn/cfps/index.htm (accessed on 30 August 2022)).

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
