# Peer review of "Exploring the Role of Land Transfer and Social Capital in Improving Agricultural Income under the Background of Rural Revitalization"

_ijerph, 2022, doi:10.3390/ijerph192417077_

Round 1
Reviewer 1 Report
The study has investigated the role of land transfer-in and social capital in improving agricultural income and the intermediary mechanism of land transfer-in. Although the research topic is very interesting, there are some problems in the research contents, the details are as follows:
1. It is recommended that the abstract of the article supplement the research methodology and expand the significance of the study, while streamlining the presentation of the study's conclusions, as the specific elaboration is already developed in the text.
2. Page 1, line 35: there is a redundant number “15”.
3. Page 2, line 93; page 3, line 110; page5, line213; page 7, line 300: the reference source is not found. Please double check and make corrections.
4. Page 3, line 115: the author(s) state that “So far, the connection between social capital and agricultural development has been paid attention by the academic circle.” Please provide at least one reference to support the statement afterwards.
5. Hypotheses are repeated in formation and listing, such repetition could be avoided by adjusting the structure of Part 2.
6. Page 14, line 549: conclusion (4) seems to be irrelevant to the main purpose of this study.
7. It is recommended that the format of the references be adjusted again.
Author Response
Dear reviewer,
Thank you very much for pointing out the flaws of the article. Now we have read and considered the article very carefully and made changed in response to your valuable comments.
Comments:
Point 1. It is recommended that the abstract of the article supplement the research methodology and expand the significance of the study, while streamlining the presentation of the study's conclusions, as the specific elaboration is already developed in the text.
Response 1: Thank you very much for your valuable comments. Follow your suggestion, we have added the relevant content in the abstract. The new abstract is as follows:
Abstract: Under the background of rural revitalization in China, with the process of urbanization and the implementation of China’s land system reform, rural workers gradually gain multiple income streams. However, increasing agricultural income remains the final guarantee for professional farmers to shake off poverty, and land is still their last security. We applied the OLS model and mediation model to a dataset of 3789 households in 25 provinces obtained from China Family Panel Studies (CFPS) to investigate the influence of farmland transfer and social capital on farmers’ agricultural incomes. The results show that farmland transfer in and social capital significantly help to increase agricultural income directly, and farmland transfer in behavior plays a vital mediating role, influencing the positive effect of social capital on agricultural income. The study examined the logical social capital-agricultural land transfer-agricultural income correlation in the progression of rural society, from "hollow" to “reflux”, under the continuous expansion of rural revitalization strategies, which is of great practical significance for re-recognizing the positive role of rural social capital and agricultural land transfer in improving the income of professional farmers and realizing the overall goal of rural revitalization. The results also provide a theoretical basis for guiding and leveraging the effective use of social capital to promote agricultural land transfer.
Point 2. Page 1, line 35: there is a redundant number “15”.
Response 2: Thank you very much for your valuable comments. We are aware of this problem, the redundant number “15” has been deleted.
Page2, line 46-48:
According to the Bulletin of the Third National Agricultural Census, by the end of 2016, large-scale businesses accounted for only 1.92% of the total agricultural operation households registered in China [1].
Point 3. Page 2, line 93; page 3, line 110; page5, line213; page 7, line 300: the reference source is not found. Please double check and make corrections.
Response 3: Thank you very much for your valuable comments. Follow your suggestions, the reference sources have been corrected.
Page 3, line 126-129:
Data from the Ministry of Agriculture and Rural Affairs of the People’s Republic of China showed that by the end of 2021, 11.2 million entrepreneurs had returned to the countryside, 1.1 million more than in2020 [17].
page 3, line 147-149:
By the end of 2019, the degree of farmland transfer exceeded one-third of the total farmland area [22].
page5, line 269-267:
Then, the expansion of cultivated land area enlarges the adjustable range of agricultural production [32].
page 7, line 358-360:
In addition, the total amount of money spent annually on relatives, neighbors, and friends, including transfers, represents the daily economic exchanges [28].
Point 4. Page 3, line 115: the author(s) state that “So far, the connection between social capital and agricultural development has been paid attention by the academic circle.” Please provide at least one reference to support the statement afterwards.
Response 4: Thank you very much for your valuable suggestions. The article provides three new references 24, 25, and 26 to support the statement.
Lines 154-158:
Thus far, connections between social capital and agricultural development have been the key focus in academia, such as the associations between different forms of social capital and innovation in agriculture [24], social capital’s critical role in agricultural and rural development in various countries [25], as well as the role of social capital in development of agricultural entrepreneurship [26].
Point 5. Hypotheses are repeated in formation and listing, such repetition could be avoided by adjusting the structure of Part 2.
Response 5: Thank you very much for your valuable comments. Follow your suggestions, the Hypotheses in listing are deleted.
Point 6. Page 14, line 549: conclusion (4) seems to be irrelevant to the main purpose of this study.
Response 6: Thank you very much for your valuable comments. Follow your suggestion, and in combination with the opinions of another reviewer’s comments, Conclusion 4 is modified as:
Lines 632-633:
(4) Farmland transfer in is conducive to increasing agricultural income, but this effect is more obvious in secondary grain-producing areas.
Point 7. It is recommended that the format of the references be adjusted again.
Response 7: Thank you very much for your valuable comments. Follow your suggestion, the format of the references has been adjusted again.
References as follows:
- Bulletin on Major Data of the Third National Agricultural Census (No.1). Available online: http://www.stats.gov.cn/tjsj./tjgb/nypcgb/qgnypcgb/201712/t20171214_1562740.html (accessed on 30 August 2020).
- Zhang J, Wan M J. Logical thinking of social capital embedded in rural revitalization and the path choice of "One Body with Two Wings". South China Agriculture 2022, 15, 239-242+252. (In Chinese) https://doi.org/10.19415/j.cnki.1673-890x.2022.15.059
- He X. Changes in the Relationship between Urban and Rural Areas and the Stage of Rural Revitalization. Guizhou Social Sciences 2021, 8, 133-138. (In Chinese) https://doi.org/10.13713/j.cnki.cssci.2021.08.019
- Zheng C. The Role of Social Capital in Social Development—Tentatively on the practice of increasing social capital in the South Korean New Village Movement. Academic Exchange 2006, 11, 127-131. (In Chinese)
- Putnam, R., Leonardi, R. & Nanetti, R. Making democracy work: Civic traditions in modern Italy. Princeton university press, Princeton, 1994.
- Xie Q. Human Capital and Social Capital: Who Can Alleviate Poverty Better? Shanghai Economic Research Journal 2017, 5, 51-60. (In Chinese) https://doi.org/10.19626/j.cnki.cn31-1163/f.2017.05.007
- Ye C, Luo L. Social Capital, Poverty Alleviation Policy and Household Welfare of the Poor-Hierarchical Linear Analysis Based on the Rural Survey Data of Guizhou Province. Finance and Economics 2011, 7, 100-109.(In Chinese)
- Lin, N. Social Capital: A Theory of Social Structure and Action. Cambridge University Press, New York, 2001. http://dx.doi.org/10.1017/CBO9780511815447
- Zhou K, He Z, WANG Z. Qu County in Shandong Province: A case Study of farmers' returning entrepreneurial behavior from the perspective of social capital. Modern Marketing (Next issue) 2022, 6, 146-148. (In Chinese) https://doi.org/10.19932/j.cnki.22-1256/F.2022.06.146
- You L, Liu J, Huo X. Aspirations, Investment and Poverty: A New Theoretical Framework of Analysis. Rural China Observations 2018, 5, 29-44. (In Chinese)
- Liu Y, Wang R. Income gap, Social Capital and Resident Poverty. Quantitative and Technical Economics 2017, 9, 75-92. (In Chinese) https://doi.org/10.13653/j.cnki.jqte.2017.09.005
- Shang H, Fan J. Research on the Role of Rural Social Organizations in Enhancing social Capital and coping with livelihood risk. Resource Development and Market 2022, 10, 1231-1237. (In Chinese)
- Li Q, Liu T, Chen Z. A Theoretical and Empirical Study on the Effect of Household Income Increase from the Perspective of Social Capital. Macroeconomic Research 2014, 1, 126-134. (In Chinese) https://doi.org/10.16304/j.cnki.11-3952/f.2014.01.011
- Liu T. Dilemma and Optimization of Rural Community Construction from the perspective of Social Capital. Journal of Xinxiang University 2022, 5, 60-63. (In Chinese)
- Zhang S, Lu M, Zhang Y. Is the role of social capital weakened or strengthened with the process of marketization? -An Empirical Study of Rural Poverty in China. Economics Quarterly (in Chinese) 2007, 2, 539-560. (In Chinese)
- Mao S. Changes and Reconstruction of Rural Social Capital in the Transition Period. Knowledge Economy 2011, 15, 67-68. (In Chinese) https://doi.org/10.15880/j.cnki.zsjj.2011.15.061
- CCTV, "Ministry of Agriculture and Rural Affairs: Chinese return to the countryside in 2021 is expected to reach 11.2 million entrepreneurs", Available online: https://baijiahao.baidu.com/s?id=1720539316790277274&wfr=spider&for=pc (accessed on 30 August 2022)
- Zhang, H. Analysis of scale operation in Chinese agricultural process: Japan's experience and enlightenment. IDE Discussion Paper, 2020, 788. Available online: http://hdl.handle.net/2344/00051725 (accessed on 30 August 2020)
- Beijing Tianze Economic Research Institute, China Land Issues Group. Land transfer and agricultural modernization. Management World 2010, 7, 66-85+97. (In Chinese) https://doi.org/10.19744/j.cnki.11-1235/f.2010.07.008
- Li, Y.; Wu, W.; Liu, Y.. Land Use Policy 2018, 74, 137-141. https://doi.org/10.1016/j.landusepol.2017.07.003
- Jiang, M.; Paudel, K.P; Mi, Y. Factors affecting agricultural land transfer-in in China: a semiparametric analysis. Applied Economics Letters 2018, 25(19-21), 1547-1551. https://doi.org/10.1080/13504851.2018.1430326
- Letter of Response to Proposal No. 01355 (No. 116 of Agricultural Water Resources) of the Fifth Session of the 13th National Committee of the Chinese People's Political Consultative Conference (Agricultural Affairs Office [2022] No. 163). Available online: http://www.moa.gov.cn/govpublic/NCJJTZ/202208/t20220831_6408228.htm(accessed on 30 August 2022).
- Sexton, R.J. Market power, misconceptions, and modern agricultural markets. American Journal of Agricultural Economics 2013, 95, 209-219. https://doi.org/10.1093/ajae/aas102
- van Rijn, Fédes & Bulte, Erwin & Adekunle, Adewale. Social capital and agricultural innovation in Sub-Saharan Africa. Agricultural Systems, Elsevier 2012, 108(C), 112-122. https://doi.org/10.1016/j.agsy.2011.12.003
- Rivera Maria, Karlheinz Knickel, José María Díaz-Puente, Ana Afonso. The role of social capital in agricultural and rural development: lessons learnt from case studies in seven countries. Sociologia Ruralis 2018, 59 (1), 66-91. https://doi.org/10.1111/soru.12218
- Hrytsaienko, M., Hrytsaienko, H., Andrieieva, L., and Boltianska, L. “The role of social capital in development of agricultural entrepreneurship,” in Modern Development Paths of Agricultural Production, ed. Nadykto; Cham: Springer, switzerland ,2019, pp. 427–440. https://doi.org/ 10.1007/978-3-030-14918-5_44
- Dingde Xu, Zhuolin Yong, Xin Deng, Linmei Zhuang and Chen Qing. Rural-Urban Migration and its Effect on Land Transfer in Rural China. Land 2020, 9(3), 1-15. https://doi.org/10.3390/land9030081
- Gao, Y.; Liu, B.; Yu, L.; et al. Social capital, land tenure and the adoption of green control techniques by family farms: Evidence from Shandong and Henan Provinces of China. Land Use Policy 2019, 89, 104250. https://doi.org/10.1016/j.landusepol.2019.104250
- Zhao H, Hua Y. Does household social capital affect rural land transfer? Land Issues Research 2021, 4, 67-75+92+183-184. (In Chinese)
- Li M, Shao B, Li Z. Study on the logical mechanism of rural revitalization and rural labor return. Journal of Xinyang Normal University (Philosophy and Social Sciences Edition) 2022, 3, 48-53. (In Chinese)
- Qian, L.; Hong, M.; Gong, L.; et al. Selection of Agricultural Land Transfer Contract from the Perspective of the Pattern of Difference Sequence and Interest Orientation. China Population, Resources and Environment 2015, 25, 95-104. (In Chinese) https://doi.org/10.13968/j.cnki.1009-9107.2015.04.007
- Liu, M.; Lu, F.; Liu, C. Study on the Influence of Farmers’ Land Transfer Behavior, Agricultural Mechanization Services on Farm Household Agricultural Income: Experience Analysis Based on CFPS 2016 Data. Nanjing Journal of Social Science 2019, 2, 26-33. (In Chinese) https://doi.org/10.15937/j.cnki.issn1001-8263.2019.02.004
- Zhang, J.; Mishra, A.K.; Zhu, P.; et al. Land rental market and agricultural labor productivity in rural China: A mediation analysis. World Development 2020, 135, 1-14. https://doi.org/10.1016/j.worlddev.2020.105089
- Liu B, Lu Q, Li X. Social capital and farm household income in poor areas-Test based on threshold regression model. Agricultural Technology and Economics 2014, 11, 40-51. (In Chinese) https://doi.org/10.13246/j.cnki.jae.2014.11.005
- Coleman, J.S. Social capital in the creation of human capital. American Journal of Sociology 1988, 94, 95-120. https://doi.org/10.1086/228943
- Kansanga, M.M. Who you know and when you plough? Social capital and agricultural mechanization under the new green revolution in Ghana. International Journal of Agricultural Sustainability 2017, 15, 708–723. https://doi.org/10.1080/14735903.2017.1399515
- Baron, R.M.; Kenny, D.A. The Moderator Mediator Variable Distinction in Social Psychological Research: Conceptual, Strategic and Statistical Considerations. Journal of Personality and Social Psychology 1986, 51(6), 1173-1182. https://doi.org/10.1037/0022-3514.51.6.1173
- Chen, H.; Wang, J. Can Social Capital Promote Farmland Transfer? A Case Study Based on China Family Panel Studies. Journal of Zhongnan University of Economics and Law 2016, 01, 21-29+158-159. (In Chinese)
- Qian, L.; Qian, W. Does Social Capital Influence Farmers' Land Transfer Behavior? —An Empirical Test Based on CFPS. Journal of Nanjing Agricultural University (Social Sciences Edition) 2017, 17, 88-89+153-154. (In Chinese)
- Yan, J.; Chen, C.; Hu, B. Farm size and production efficiency in Chinese agriculture: output and profit. China Agricultural Economic Review 2019, 11, 20-38. https://doi.org/10.1108/CAER-05-2018-0082
Thank you very much for pointing out the flaws of the article again!
Best wishes!

Reviewer 2 Report
The paper investigates the impact of social capital on agricultural income.
Some questions should be elaborated in the introduction section. What is the structure of social capital before rural revitalization? How does social capital change due to rural revitalization? What do you mean by new social capital?
The term “Land transfer-in” should be replaced by another term that is more familiar in Western literature. If the authors create the term, it should be clearly explained in the earlier part of the paper.
To my understanding, social capital does not have direct relationship with agricultural income. Instead, social capital affects income through “land transfer-in.” If this is true, the authors should clarify their relationship throughout the writing.
A few comments are in the PDF file.
I suggest the paper be professionally edited. I caught a few minor errors which I mark in the PDF file.

Author Response
Dear reviewer,
Thank you very much for pointing out the flaws of the article. Now we have read and considered the article very carefully, and made changed in response to your valuable comments.
Comments:
Point 1. Some questions should be elaborated in the introduction section. What is the structure of social capital before rural revitalization? How does social capital change due to rural revitalization? What do you mean by new social capital?
Response 1:Thank you very much for your valuable comments. Follow your suggestion, the questions you raised are elaborated in the introduction section.
Lines 81-115:
In the contemporary period, social capital is still playing a role in promoting farmers' income. Under rural revitalization strategies and land system reforms, the mobility of rural residents has greatly increased, leading to gradual loosening of the interpersonal network; thus, the countryside is no longer a closed community [14], reshaping traditional rural social capital which is mainly characterized by acquaintances and closed networks. In the process of agricultural modernization, gradual improvements in the market system and the development of rural areas from closed to open are widely believed to have eroded the "Chinese guanxi" basis of the role of traditional rural social capital, which has led to doubts about the role of social capital in promoting farmers' income. For example, Zhang Shuang et al. hypothesized that with improvements in marketization, the role of social capital in poverty will be weakened, and especially the role of family social networks [15].However, it should also be observable that the urban-rural relationship has smoothly undergone the rural "hollowing out" stage and entered the return stage of migrant workers, which has brought back talent, technology, and capital. At this time, the network of new social capital is more open and contains a wider range of social relationships, including more urban relationships. On the normative side, the digital economy and the Internet have prompted social capital to act in a more resilient manner. Additionally, the Internet has further reduced constraints on the level of trust, improved the gray character which traditional social capital possesses, and expanded the possible ways in which social capital can work to boost farmers' incomes and fight poverty. Under rural revitalization, novel rural social capital is becoming more modern, advanced, extensive, and digital. As a kind of invisible "soft" capital, it plays a wider role in attracting human capital, material, and financial resources, forming a good circulation network between rural and urban areas, and reinforcing mechanisms of the role of traditional social capital in promoting farmers' agricultural income, i.e., the expansion of farmers' financial sources, the enrichment of information, and the establishment of social networks. Therefore, social capital and rural revitalization are mutually supportive. The embedding of social capital is conducive to the realization of rural revitalization. The implementation of the rural revitalization strategy enriches rural social capital and attracts more social capital back to the countryside [2]. Moreover, the degree of modernization in rural areas in China is weak in the transition period. Under the condition that "system" and "market" are not sound, "relationship" plays an extremely important role as one of the bases and ways for farmers to obtain scarce resources [16].
Point 2. The term “Land transfer-in” should be replaced by another term that is more familiar in Western literature. If the authors create the term, it should be clearly explained in the earlier part of the paper.
Response 2: Thank you very much for your valuable comments. Follow your suggestions, the term “Land transfer-in” is replaced by another term “Land transfer in”, which comes from the article of scholars Xu et al (2020), and it is clearly explained in the introduction section Page 4, line 169-170: Land transfer in means that farmers rent land from other farmers in the village through contract or oral contract.
Line 168-176:
Land transfer is measured by two variables: land transfer in and land transfer out. Land transfer in means that farmers rent land from other local farmers through written or verbal contract [27]. It is more directly related to farmers' agricultural income, and was the research object of this study. China is a society of traditional “Chinese guanxi”, in which social capital plays a supplemental role in the formal system [28]. In fact, rural households with more capital can more easily transfer into land for large-scale production, and thus earn more farm income [29]. Land transfer is an important bridge for social capital to embed in rural revitalization, which ultimately raises farming income for farmers.
Points 3. To my understanding, social capital does not have direct relationship with agricultural income. Instead, social capital affects income through “land transfer-in.” If this is true, the authors should clarify their relationship throughout the writing.
Response 3: Thank you very much for your valuable comments. Follow your suggestions, their relationship clarifications have been added throughout the text. The empirical test results show that social capital has direct relationship with agricultural income and the result is robust, and social capital affects income through “land transfer in” is true.
Lines 27-30:
The results show that farmland transfer in and social capital significantly help to increase agricultural income directly, and farmland transfer in behavior plays a vital mediating role, influencing the positive effect of social capital on agricultural income.
Lines 207-209:
Thus, there is also a direct impact of social capital on the agricultural income of general agricultural workers, who are the largest population of the most disadvantaged in society.
Lines 244-247:
H1. Rural social capital can still contribute to the growth of agricultural income of farm households through various mechanisms, i.e., social capital has direct relationship with agricultural income.
Lines 282-292:
2.3. Impacts of Social Capital on Farmland Transfer in Behaviors
The role of social capital in all aspects of agricultural production is similar. The process of land transfer for agricultural income generation, involves a reduction in transaction costs, the access to lease capital and information, risk sharing and cooperation mechanisms in the establishment of long-term land transfer behavior, and the input of urban production factors, all carried out under a network of rural social. Therefore, rural social capital can effectively promote farmers' land transfer in behavior, which, in turn, promotes agricultural income growth. Thus, land transfer plays an intermediary role in the process of social capital promoting the growth of farmers' income and is one of the mechanisms through which social capital directly affects farmers' agricultural income. Accordingly, research hypothesis 3 was formulated:
H3: Social capital has a positive effect on land transfer in, and land transfer in is an important way for social capital to increase agricultural income, which plays an intermediary role.
Lines 560-565:
- Discussion
Based on the data of 3789 rural households from China Family Panel Study, this study examined the direct impact of farmland transfer in and social capital on agricultural income, and the mediating role of land transfer in in the process of social capital for farm income generation.
Lines 614-633:
- Conclusions
Based on this analysis, several conclusions can be drawn.
(1) In rural China, social capital is a useful resource for farmers, which is conducive to farmland transfer in, thus contributing to increases in agricultural income. With the increases in social capital investment, agricultural income will also increase, and after controlling other related variables, the marginal contribution of social capital investment to agricultural income is 16.56% and the marginal effect of farmland transfer in on agricultural income is 1.19. Social capital and land transfer in are directly related to agricultural income. In addition, the impact of social capital and farmland transfer in on agricultural income presents individual heterogeneity and regional heterogeneity.
(2) The farmland transfer in is a mediating variable, which is one of the important channels for social capital to influence agricultural income. Land is a direct factor of agricultural income and one of the mechanisms by which other variables affect agricultural income.
(3) Farmland transfer in ratio is a threshold variable, and under the constraint of the scale of farmland transfer in, a nonlinear relationship exists between social capital and agricultural income.
(4) Farmland transfer in is conducive to increasing agricultural income, but this effect is more obvious in secondary grain-producing areas.
Point 4. A few comments are in the PDF file. I suggest the paper be professionally edited. I caught a few minor errors which I mark in the PDF file.
Response 4: Thank you very much for your valuable comments. The minor errors which you mark in the PDF file have been corrected, and we have made extensive English revisions to the manuscript through the paid editing service. Thank you again!
Line 44-45:
The issue of absolute poverty in China was solved historically; however, there will still be people earning low incomes in society.
Lines 60-62:
In China, personal relationships are extremely important in society, thus, social capital exists among family, kinship, and neighborhood relationships.
Lines 81-86:
In the contemporary period, social capital is still playing a role in promoting farmers' income. Under rural revitalization strategies and land system reforms, the mobility of rural residents has greatly increased, leading to gradual loosening of the interpersonal network; thus, the countryside is no longer a closed community [14], reshaping traditional rural social capital which is mainly characterized by acquaintances and closed networks.
Lines 117-153:
Under rural revitalization strategies, local employment remains the most effective and sustainable means of alleviating poverty, considering the long-term effectiveness and cost. The cyclical opening and closing of cities caused by the COVID-19 pandemic has led to a decline in employment demand in some urban industries, especially in the contact service industry, ultimately leading to a substantial increase in migrant workers returning home for employment. Data from the Ministry of Agriculture and Rural Affairs of the People’s Republic of China showed that by the end of 2021, 11.2 million entrepreneurs had returned to the countryside, 1.1 million more than in 2020 [17]. Therefore, the local employment of migrant workers urgently needs to promote the large-scale development of agricultural industrialization. Land is the most basic and important production factor for agricultural-scale management; moderate-scale agriculture relies on the consolidation of agricultural land and farmland transfer [18]. Thus, an increase in land transfer provides an important guarantee for agricultural modernization [19]. In rural China, the Household Responsibility System, implemented in the early 1980s, promoted production motivation in peasants, and led to serious farmland fragmentation, which curbed large-scale agriculture operation [20]. In 1988, the transfer of land contract management rights was officially permitted [21], which effectively promoted the transfer of farmland and created better conditions for moderate-scale agricultural management. By the end of 2019, the degree of farmland transfer exceeded one-third of the total farmland area [22]. For professional farmers, one of the best ways to increase income is by transferring into land and developing it into a large agricultural enterprise. In order to improve the efficiency of resource allocation, professional farmers should expand the scale of farmland to cater to contemporary agricultural production [23].
Lines 205-207:
Social capital can effectively reduce the incidence of poverty and the income gap through a variety of ways, which has been confirmed by many scholars [2,6,11,13].
Thank you very much for pointing out the flaws of the article again!
Best wishes!

Round 2
Reviewer 2 Report
I have no further comments.